# A Double Histochemical/Immunohistochemical Staining for the Identification of Canine Mast Cells in Light Microscopy

**DOI:** 10.3390/vetsci8100229

**Published:** 2021-10-12

**Authors:** Francesca Gobbo, Giuseppe Sarli, Margherita De Silva, Giorgia Galiazzo, Roberto Chiocchetti, Maria Morini

**Affiliations:** Department of Veterinary Medical Sciences, University of Bologna, Ozzano dell’Emilia, 40064 Bologna, Italy; francesca.gobbo3@unibo.it (F.G.); giuseppe.sarli@unibo.it (G.S.); margherita.desilva2@unibo.it (M.D.S.); giorgia.galiazzo2@unibo.it (G.G.); roberto.chiocchetti@unibo.it (R.C.)

**Keywords:** canine mast cell tumors, co-localization, immunohistochemistry, toluidine blue stain

## Abstract

Immunohistochemistry (IHC) is a widely used technique in diagnostic pathology, but the simultaneous analysis of more than one antibody at a time with different chromogens is rather complex, time-consuming, and quite expensive. In order to facilitate the identification of mast cells (MCs) during immunohistochemical analysis of membrane and/or nuclear markers, we propose a new staining method that includes the association of IHC and toluidine blue as a counterstain. To achieve this goal, we tested c-kit, Ki67, and cannabinoid receptor 2 on several cases of cutaneous canine mast cell tumors (MCTs), cutaneous mastocytosis, and atopic dermatitis. The results obtained show how this double staining technique, although limited to non-cytoplasmic markers and of little use in poorly differentiated MCTs in which MC metachromasia is hard to see, can be used during the evaluation of nuclear and/or membranous immunohistochemical markers in all canine cutaneous disorders, especially if characterized by the presence of a low number of MCs. It can help to evaluate those MCTs in which neoplastic MCs must be clearly distinguished from inflammatory cells that can infiltrate the tumor itself, in facilitating the calculation of the Ki67 index. Moreover, it can be used to study the expression of new markers in both animal and human tissues containing MCs and in MC disorders.

## 1. Introduction

In diagnostic pathology, immunohistochemistry (IHC) is a technique used for the identification of cellular markers in tissue samples that could influence disease diagnosis and, therefore, patient management [1]. To fully understand the different pathologies, it is also important to identify which cells specifically express markers or receptors, in order to identify new target cells for new therapies.

Mast cells (MCs) are investigated in several types of pathologies in veterinary medicine, such as canine atopic dermatitis (AD), cutaneous mastocytosis, and cutaneous mast cell tumors (MCTs) [2,3]. A hallmark of their identification is the metachromatic red/violet color of the cytoplasmic granules [4] that follows some staining, such as toluidine blue, May-Grünwald Giemsa, and Leishman. This staining allows the detection of MCs in tissues and is strongly recommended as a routine stain for this purpose [4]. However, in immunohistochemical investigations aimed at demonstrating antigen expression in normal or neoplastic MCs, a limit exists with the irrefutable demonstration, in that the investigated antigen is expressed in MCs. By IHC, the objective way to demonstrate a mast cell phenotype is the cytoplasmic presence of the constituents of its secretory granules, such as chymase and/or tryptase [5,6,7,8]. Tryptase and chymase are both used in the detection of normal [8,9] or neoplastic [10,11,12] conditions, and chymase activity is closely correlated with inflammatory diseases progression [8].

Atopic dermatitis in dogs is a chronic inflammatory skin disease involving abnormalities in skin barrier function and cutaneous inflammation [13]. In canine AD, the skin perivascular infiltrate is mixed, composed mostly of T cells, dendritic cells, eosinophils, and MCs that appear to be increased in number compared to non-pathological canine skin [14].

Cutaneous mastocytosis, which resembles a subset of urticaria pigmentosa (UP) in humans, is a rare disease in dogs [15,16]. The lesions caused by UP are characterized by infiltrate of fairly monomorphic, well-differentiated MCs that may extend from the superficial to the deep dermis [7]. In humans, some forms of cutaneous mastocytosis are associated with mutation of the c-kit oncogene in MCs, and a study has proven a mutation of the exon 11 of c-kit in canine cutaneous mastocytosis [17].

MCTs are the second most frequently diagnosed malignancy in dogs [18], and canine cutaneous MCTs account for approximately 20% of all canine skin tumors [19]. The current prognostic factors of MCT include histological criteria considered for the three-tier Patnaik grading system (grades 1, 2, and 3) [20] and the two-tier Kiupel grading system (low and high grade) [21], aberrant expression pattern of c-kit [22], gain-of-functions mutations in c-KIT oncogene [23,24], and Ki67 index [25]. C-kit, also known as mast/stem cell factor receptor (SCFR) or CD117, is a trans-membrane type III tyrosine kinase receptor encoded by the c-Kit proto-oncogene [26]. Its expression has previously been demonstrated by IHC in normal and neoplastic canine MCs [27,28]. In canine MCTs, its expression could be membranous, cytoplasmic, or focal cytoplasmic (Golgi-like), and its type of expression correlates with different histological parameters [29,30,31,32]. Proliferation markers are commonly used for the prognostication of cutaneous mast cells tumors, and one of the most ordinarily used in veterinary medicine is Ki67 [33], which is exclusively detected within the nucleus during the interphase [34].

Cannabinoid receptor 2 (CB2R) is the endocannabinoid subtype 2 receptor, which is expressed in the nucleus and cytoplasm of many cells [35,36]. In the canine gastrointestinal tract, cells expressing CB2R are lamina propria cells, MCs, immunocytes, endothelial cells, and smooth muscle cells [37]. In the skin of healthy dogs and of dogs with AD CB2R, immunoreactivity is expressed in perivascular cells with MC morphology, fibroblasts, and endothelial cells [19], suggesting a potentially therapeutic target role of CB2R in the treatment of AD and other inflammatory diseases in companion animals.

During histologic examination of MCTs, AD, and mastocytosis, it could be useful to determine the presence of MCs within the tissue, and the co-expression of different markers can be detected by multiplex immunohistochemistry or immunofluorescent analysis [38]. These techniques allow the identification of the co-expression of many markers simultaneously and it is the key technique for visualizing the organization of endogenous cellular structures in single cells [39,40]. Unfortunately, those methods are quite complex and expensive, time-consuming, and quite difficult to interpret, and immunofluorescence staining requires the use of an expensive fluorescence microscope, not in use in most diagnostic laboratories.

Herein, we wanted to test and prove the effectiveness of a modified counterstain applied in immunohistochemistry when MC identification is difficult and may be necessary to objectivate the expression of different immunohistochemical markers or receptors in MCs. The method consists of the combination of immunohistochemistry followed by toluidine blue (TB) used as counterstain. Via this method, we analyzed various canine MC disorders using different markers known to have prognostic and therapeutic implications. The aim of this study was to verify the validity of the new method and to suggest its practical use, as well as its future applications with other different nuclear and/or membranous immunohistochemical markers.

## 2. Materials and Methods

### 2.1. Sample Collection

We selected, from the Archive of the Pathology Service of the Department of Veterinary Medical Sciences (University of Bologna, Ozzano dell’Emilia, Bologna, Italy), tissues/specimens from 17 canine cutaneous MCTs (previously classified according to the WHO histological classification of domestic animals [32]), four AD, and two cutaneous mastocytosis already diagnosed according to Gross et al. in 2005 [7].

All MCTs were previously histologically graded following both the Patnaik histologic grading system [20] (six Patnaik grade 1, four Patnaik grade 2, and seven Patnaik grade 3) and the Kiupel system [21] (10 Kiupel low grade and seven Kiupel high grade).

### 2.2. Immunohistochemistry and Toluidine Blue Counterstain

We performed immunohistochemical analysis of c-kit, Ki67, and CB2R of all selected cases, and TB was used as a counterstain instead of the routine Harris hematoxylin (HH) stain.

Two micron-thick sections of each sample were dewaxed and rehydrated. Endogenous peroxidase was blocked by immersion in 0.3% H_2_O_2_ in methanol for 30 min at room temperature (RT). The primary antibodies, dilutions, antigen retrieval methods, and tissues used as positive controls are reported in Table 1. Antigen retrieval was followed by cooling at RT for 20′. Blocking of non-specific antigenic sites was achieved by incubating the slides in a solution of 10% normal goat serum in PBS for 30′ at RT and afterward incubated overnight at 4 °C with the primary antibodies.

Binding sites were revealed by secondary biotinylated antibody and amplified using a commercial avidin–biotin–peroxidase kit (ABC Kit Elite, Vector, Burlingame, CA, USA). The chromogen 3,3′-diaminobenzidine (0.05%) (Histo-Line Laboratories, Emergo, Europe) was used. Slides were counterstained by immersion in toluidine blue (0.03%) (MERCK and Co., Readington, NJ, USA) for 7 min and then washed in tap water for 5 min. Sections were dehydrated briefly (10–20 s each passage) in Ethanol 80, 95, 100, 100 and xylene and permanently mounted with DPX medium (Fluka, Riedel-de Haen, Germany).

To compare the differences between the Harris hematoxylin and toluidine blue counterstains, we performed c-kit and Ki67 IHC of MCTs of different Patnaik and Kiupel grades, and HH was used as the counterstain following the standard procedures.

## 3. Results

All of the cases tested were evaluable and included in the study.

C-kit immunohistochemical analysis of canine MCTs showed the three different patterns, and we found that the metachromatic reaction was clearly detectable in all cases of membranous and Golgi-like c-kit expression (Figure 1B,C and Figure 2C); by contrast, the metachromatic reaction can be hidden from the immunohistochemical reaction of c-kit in the cytoplasm (Figure 1A)—in our cases observed in poorly differentiated MCTs (Patnaik grade III, Kiupel high grade) (Figure 3F). Mast cells within the dermis in AD showed cytoplasmic and membranous expression of c-kit (Figure 2A), similarly to the well-differentiated MCs of mastocytosis (Figure 2B).

During the evaluation of the IHC of Ki67, in all canine MCTs and mastocytosis, we were able to see the cytoplasmic metachromatic reaction of the granules and the nuclear immunohistochemical expression of Ki67 in the same cells (Figure 2D–F). By contrast, in AD, Ki67 was positive only in the epithelial cells of hair follicles; no mast cells showed a state of active proliferation.

The immunohistochemical analysis of CB2R revealed nuclear expression of the receptor in MCs of mastocytosis (Figure 2H) and of MCTs (Figure 2I), and then the metachromatic reaction of CB2R expressing cells was clearly detectable. By contrast, in AD, MCs expressed cytoplasmic CB2R immunolabeling (Figure 2G), then the metachromatic reaction was hidden by the chromogen used in IHC.

Comparison between the HH and TB counterstains shown in Figure 3. A membranous (Figure 3A,D) and Golgi-like (Figure 3B,E) KIT pattern of MCTs Patnaik grade 1 (Figure 3A,D) and grade 2 (Figure 3B,E) or Kiupel low grade (Figure 3A,B,D,E) is clearly visible, either by observing IHC counterstained with HH or TB. Mast cell tumors Patnaik grade 3 or Kiupel high grade showed cytoplasmic c-kit expression that hid the metachromatic reaction of TB *(*Figure 3C), and a subset of MCs did not show c-kit expression, instead showing only the metachromasia (Figure 3F).

By Ki-67 immunohistochemical analysis, we observed only few MCs positive for Ki67 on MCTs Patnaik grade 1 and 2 or Kiupel low grade (Figure 3G,H,J,K); by contrast, MCTs Patnaik grade 3 or Kiupel high grade showed a high proliferative status (Figure 3I,L). A comparison between the HH and FTB counterstains revealed the clear detection of MCs within the tissue if counterstaining was performed by TB (Figure 3G–L).

## 4. Discussion

The most common way to demonstrate the expression of antigens in MCs is to perform a double immunofluorescent analysis [39], including an anti-tryptase and/or anti-chymase antibody to provide obvious evidence of MCs. The human and veterinary literature lacks other types of histochemical approaches than HH to counterstain IHC. Other stains aim to visualize MCs during histochemical evaluation in light microscopy, where MCs are usually identified by their morphology. Since, by HH, the intracytoplasmic granules of MCs are not highlighted, it is difficult to differentiate these cells from other types of round cells (plasma cells and histiocytes, for example).

Herein, we illustrated and tested a new proposal for a staining method that combines immunohistochemistry with a specific histochemical counterstain for MCs. This approach is useful for proving the expression of nuclear and membranous markers on MCs, but not of cytoplasmic markers, because the presence of chromogen in the cytoplasm can hide the metachromatic staining. Even if this limit does not seem to have been solved, the proposed method is a valid alternative to immunofluorescence to demonstrate nuclear or membranous antigens in MCs, being simple to perform and very inexpensive.

In dogs, MCTs are the second most frequently diagnosed malignancy of MCs [42] and belong to the round-cell tumors of the connective tissue [43], which also includes histiocytomas, lymphomas, plasmacytomas, and transmissible venereal tumors. Then, in some cases, immunohistochemical investigation of membrane (lymphocyte or histiocytes markers) or nuclear (MUM-1 (MUltiple Myeloma 1) for plasma cells) antigens are required for diagnosis [44,45]. Therefore, the use of the technique we propose could have the advantage, when the cytoplasmic metachromasia is present, to quickly direct to a tumor diagnosis of MCTs performing a single test. We are aware that poorly differentiated MCTs do not always demonstrate the metachromasia characteristic of TB staining [5,46]. This could be a limit of the method, although in our study, all MCTs of Patnaik grade 3 or Kiupel high degree still showed evident metachromasia (Figure 3F,L).

Mast cell tumors can be heterogeneous in clinical presentation and biological behavior [47]. In MCT evaluation, the Ki67 index is used as a prognostic factor of recurrences [18] and survival [23]. The quantitation of the percentage of Ki67 positive nuclei, especially when MCTs present a severe amount of inflammatory infiltrate, could be easier and more truthful using this double histochemical staining, because it allows the assessment of all cells with a metachromatic cytoplasmic stain indicative of MCT cells. During immunohistochemical evaluation of the c-kit pattern, which correlates with histological grade and post-surgical prognosis [29], this method can help in case of membranous or Golgi-like expression (KIT-pattern I and II), as observed in this study; on the other hand, it is difficult to evaluate the cytoplasmic c-kit pattern present on poorly differentiated MCTs with a high degree of malignancy (KIT pattern III). Moreover, this staining technique can also be used during the evaluation of other nuclear markers in canine MCTs, such as p21 [48], p27 [48], p53 [49,50], and Mdm2 [49].

In canine cutaneous disorders such as AD and mastocytosis, MCs can be a key diagnostic element [14], but unfortunately, our results showed that this technique is not usable for c-kit assessment, as non-neoplastic MCs show, in addition to membrane positivity, strong cytoplasmic positivity; however, it is practicable in the case of the immunohistochemical study of nuclear and membranous markers. Therefore, it could be interesting and helpful to visualize MCs in order to find new membranous and/or nuclear markers with potential therapeutic implication, such as CB2R [19,36].

## 5. Conclusions

The proposed method, which combines IHC and histochemistry to exploit the typical cytoplasmic metachromasia of the MCs, could be an effective and easily practicable method facilitating the identification of MCs in the context of immunohistochemical analysis with different membrane and/or nuclear markers, and it could find functional use for future applications in veterinary and human tissue. Therefore, although limited to the assessment of non-cytoplasmic antigens, it can be considered a valid alternative to multiplex IHC or double immunofluorescence techniques.

Nevertheless, further evidence with a higher number of cases and with additional markers is needed to expand the usefulness of this technique.

## Figures and Tables

**Figure 1 vetsci-08-00229-f001:**
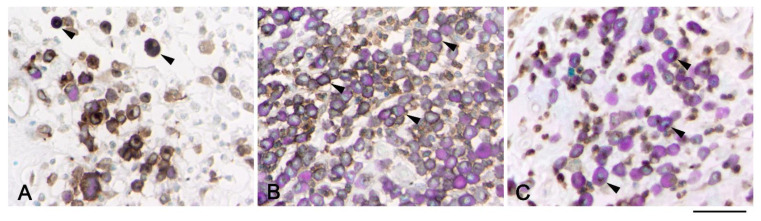
Association between toluidine blue stain and immunohistochemistry of c-kit. Canine mast cell tumors showing cytoplasmic (**A**, arrows), membranous (**B**, arrows), and Golgi-like (**C**, arrows) expression of c-kit. Metachromatic red stain in the cytoplasm is present in (**B**) (membranous) and (**C**) focal cytoplasmic type of expression, while it is hidden in (**A**) when a cytoplasmic diffuse immunohistochemical stain is present. Scale bar = 50 µm.

**Figure 2 vetsci-08-00229-f002:**
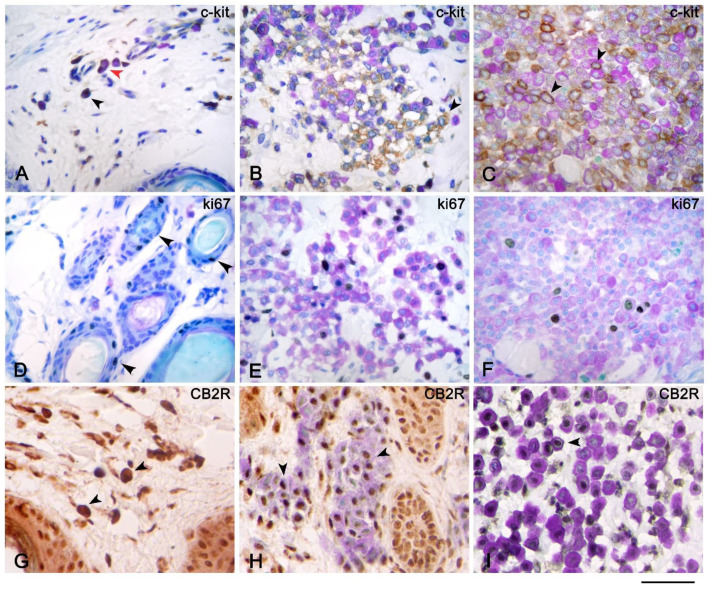
Immunohistochemistry of c-kit, Ki67, and CB2 receptor with toluidine blue as the counterstain of atopic dermatitis (AD), cutaneous mastocytosis, and cutaneous mast cell tumors (MCTs). Immunohistochemistry of c-kit of AD (**A**), mastocytosis (**B**), and MCTs (**C**) showed cytoplasmic (black arrow) and membranous (red arrow) (**A**,**B**) and only membranous (black arrow) (**C**) c-kit expression of mast cells (MCs). Immunohistochemistry of Ki67 of mastocytosis (**E**) and MCTs (**F**) showed nuclear expression of MCs (**E**,**F**) visualized by toluidine blue (TB) stain, and MCs in AD were not in cycle; therefore, Ki67 stain was found only on the epithelial cells of hair follicles (black arrows) (**D**). Mast cells of mastocytosis (**H**) and MCTs (**I**) revealed nuclear expression of CB2 receptor (black arrows); by contrast, the MCs of AD (**G**) showed cytoplasmic expression of the CB2 receptor (black arrow) that hid the metachromatic reaction of TB stain. Scale bar = 50 µm.

**Figure 3 vetsci-08-00229-f003:**
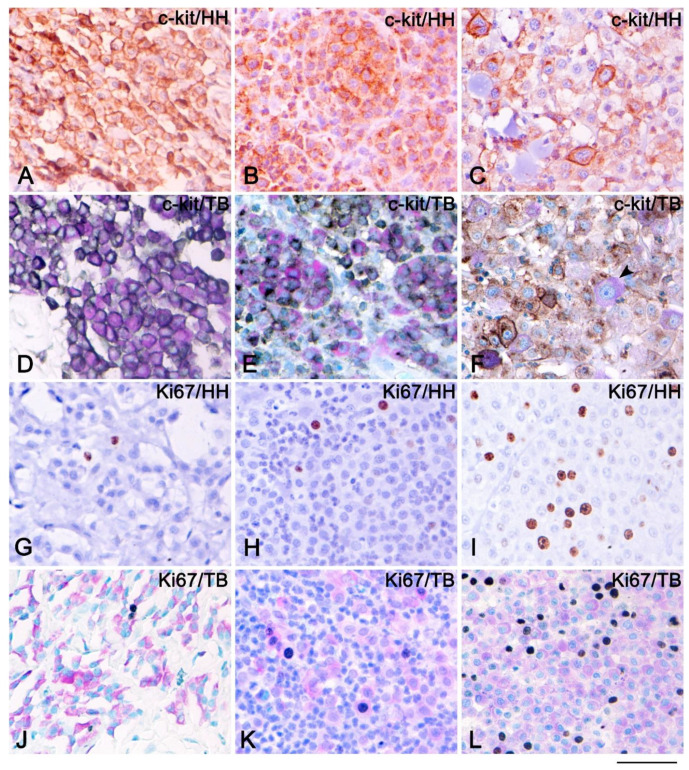
Immunohistochemistry of c-kit and Ki67 counterstained either with Harris hematoxylin (HH) or toluidine blue (TB) stain of cutaneous mast cell tumors (MCTs). Immunohistochemistry of c-kit of MCTs Patnaik grade 1 and Kiupel low grade counterstained either with HH (**A**) or TB (**D**) showed membranous expression of c-kit; therefore, the metachromatic reaction of the TB stain is visible. Immunohistochemistry of c-kit of MCTs Patnaik grade 2 and Kiupel low grade counterstained either with HH (**B**) or TB (**E**) showed a membranous and Golgi-like pattern of c-kit; therefore, the metachromatic reaction of TB is detectable. Immunohistochemistry of c-kit of MCTs Patnaik grade 3 and Kiupel high grade counterstained either with HH (**C**) or TB (**F**) showed a cytoplasmic pattern of c-kit, and some MCs were negative for c-kit (black arrow); therefore, the metachromatic reaction is visible only in c-kit negative MCs. Immunohistochemistry of Ki67 of MCT Patnaik grade 1 (**G**,**J**), grade 2 (**H**,**K**), and grade 3 (**I**,**L**) counterstained either with HH (**G**–**I**) or TB (**J**–**L**). Scale bar = 50 µm.

**Table 1 vetsci-08-00229-t001:** Immunohistochemistry materials and methods. Abbreviations: MW, microwave; ON, overnight; Ag, antigen [Reference of cross-reactivity with canine MCs/canine tissues].

Marker	Type, Clone, Reference	Supplier	Dilution/Incubation	Ag Retrieval
c-kit (CD117)	Rabbit polyclonal anti-c-kit (A4502) [41]	Dako, Glostrup, Denmark	1:300 ON	10′ Citrate pH6 MW: 750 W
Ki67	Mouse monoclonal anti-Ki67 (MIB1) [41]	Dako, Glostrup, Denmark	1:600 ON	20′ Citrate pH6 MW: 750 W
CB2R	Rabbit polyclonal anti-Cannabinoid Receptor II (ab45942) [37]	Abcam, Cambridge, UK	1:500 ON	10′ EDTA pH8 MW: 750 W

## Data Availability

The data presented in this study are available on request from the corresponding author.

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
