# Peer review of "A Double Histochemical/Immunohistochemical Staining for the Identification of Canine Mast Cells in Light Microscopy"

_vetsci, 2021, doi:10.3390/vetsci8100229_

Round 1

Reviewer 1 Report

Although, mast cell tumor is a common disease in dogs, knowing a new technique for the identification of mastocytoma using IHC is useful. For this purpose, the authors tried to use a TB staining instead of routine hematoxyline staining as a counterstain of IHC. The result was shown in Figs 1 and 2. By seeing the figures, it is understood that the strong cytoplastic IHC staining wipes out the metachromatic TB staining but the nuclear or membranous staining does not. Although no diagnostic utility of TB staining is shown.

Figures of IHC staining with hematoxyline and TB counterstaining should compared and the superior aspects of TB staining should clearly demonstrated.

Reviewer 2 Report

Dear authors,

I understand the idea to combine touluidin blue stain and immunohistochemistry. However, I have some critical points:

  • for double immunohistostains are further colours available than red and brown. But I agree, flouresecence is more comfortable than immunohisto-double labelling. However, I think this should not take so much part in the abstract
  • Explanation of the markers you use in this study (cKIT, Ki67, CB2R should not appear in results but in the introduction. Especially the diagnostic and prognostic value of CB2R detection should be explained because this is not commonly used. --> introduction should be improved
  • The MCT which were used in this study should be characterized in more detail (graduation according to Patnaik et al. and Kiupal et al.)
  • There are only few cases - howe were thwey selected? According to the different cKIT expression patterns?
  • as you did all methods on 4MCT, 2 CM and 4AD you should not repeat it always in the results...
  • In the poorly differentiated mast cell tumors, there are few granules to be seen, so that the double staining cannot be helpful at all! The inner logic of the argument cannot be seen here! You did not describe your MCT in deteial - especially their touidinblue staining intensity
  • the pictures of figure 1 are not in focus
  • and it appears that the toluidin stain is overlapping the immunostaining thus there is no effort in doing double staining
  • the same problme in Fib 2A-C ckit there seem to be a lot of cells only be stained by toluidin but cKit expression is not recognizable
  • The idea to combine Ki67 and touidin stain may be helpfull in markedly inflamed MCT as only mastcells should be evaluated for prognostic evaluation according to Webster et al. However from your pictures, it appears that cell in mitotic cycle are poorly stained with Toluidin blue. However, evaluation of Ki67 for prognostzic purposes is only sensefull in the tumour not in the metastases.....
  • In picture 2I there so many nuclei unlabelled?
  • I would prefer in general that a larger series of cases should be investigated to discuss the advantages and limitations of this method in MCT of different degrees
  • In conclusion, I think this was a nice idea but study design and critical interpretation of the result should be improved markedly for the diagnostic and prognostic relevance

Reviewer 3 Report

I believe the paper is worthy of publishing as a technical note rather than a research article.

The work is original and represents a proof of principle study on the use of toluidine as a counterstain for increasing the specificity of detection of neoplastic mast cells in combination with immunohistochemical markers.

The abstract and the discussion needs to emphasise its originality as the literature curiously does not seem to cite any similar use of toluidine blue in the context of immunohistochemical detection of mast cells in either human or animal tissue.

Round 2

Reviewer 1 Report

The authors have revised according to the reviewer's comments. I have no further comments.

Reviewer 3 Report

I have read the new version with great interest, and I am pleased to report that the original manuscript (version V1) has been considerably improved and the version is acceptable in form of an original research article as opposed to previously recommended technical note. The authors have included considerable amount of new material to substantiate their claims about the usefulness of the novel approach of adopting toluidine blue as a counterstain instead of the conventional Haematoxilyn and Eosin, for immunohistochemical analysis of mast cell lesions associated with animal tissue. The manuscript is well written and is illustrated and supported with excellent quality photomicrographic images.